# A Workflow for Uncertainty Assessment in Elemental Analysis of Archaeological Ceramics: A Case Study of Neolithic Coarse Pottery from Eastern Siberia

Galina V. Pashkova [1,*], Mikhail A. Statkus [2], Maria M. Mukhamedova [1,3], Alexander L. Finkelshtein [3,4], Irina V. Abdrashitova [2], Olga Yu. Belozerova [4], Victor M. Chubarov [1,4], Alena A. Amosova [4], Artem S. Maltsev [1], Elena I. Demonterova [1] and Dmitriy L. Shergin [3]

[1] Institute of the Earth's Crust, SB RAS, 128 Lermontov St., 640033 Irkutsk, Russia; m.mukhamedova2017@yandex.ru (M.M.M.); chubarov@igc.irk.ru (V.M.C.); maltsev@crust.irk.ru (A.S.M.); dem@crust.irk.ru (E.I.D.)

[2] Chemistry Department, Lomonosov Moscow State University, 1-3 Leninskie Gory, 119991 Moscow, Russia; statkusma@my.msu.ru (M.A.S.); 1543irina@rambler.ru (I.V.A.)

[3] Pedagogical Institute, Irkutsk State University, 1 K. Marx St., 664003 Irkutsk, Russia; finkel@igc.irk.ru (A.L.F.); dmitriy-shergin@rambler.ru (D.L.S.)

[4] Vinogradov Institute of Geochemistry, SB RAS, 1 A. Favorsky St., 664033 Irkutsk, Russia; obel@igc.irk.ru (O.Y.B.); amosova@igc.irk.ru (A.A.A.)

* Correspondence: pashkova.gv@yandex.ru

**Abstract:** In this study, the assessment of uncertainties introduced at different stages of the elemental analysis of archaeological ceramics has been described using the example of the Neolithic pottery sherds from Popovsky Lug (eastern Siberia). To evaluate the uncertainty introduced by sampling due to ceramic heterogeneity, three original sherds were cut into small subsamples. Powdered subsamples (250–350 mg) were analyzed using wavelength-dispersive X-ray fluorescence and inductively coupled plasma mass spectrometry methods, and the variations between analytical results for independent subsamples were compared with the variations introduced during the analytical process (measurement and sample preparation). It was shown that 250–350 mg of ceramic is sufficient to obtain good reproducibility (2–15%) between separate subsamples for most major and trace elements, even for the heterogeneous Neolithic ceramics included in this study. The differing behavior of concentration variations in some elements was explained by measuring the ceramic cross-sections by scanning electron microscopy and micro-X-ray fluorescence spectrometry, as well as by the theoretic modeling of the sampling error. The described workflow can be useful in finding uncertainties in elemental analysis results, which may affect the interpretation of bulk chemical composition in ceramic provenance studies.

**Keywords:** archaeological ceramics; elemental analysis; WDXRF; ICP-MS; SEM; μXRF

## 1. Introduction

Data on the elemental composition of ancient ceramics are often used in archaeological provenance studies for the identification of pottery from different regions, characterization of raw materials, and manufacturing processes [1–3]. Ceramic is among the most complex archeological materials because it consists of clay minerals, non-plastic inclusions, and intentionally added tempers in different proportions. For archaeological ceramics, quantitative elemental analysis can be performed by means of analyzing dry, powdered, and homogenized ceramic fragments using instrumental neutron activation analysis (INAA) [4–6], inductively coupled plasma mass spectrometry (ICP-MS) [7–9], conventional X-ray fluorescence spectrometry (XRF) [7,10–14], and total reflection XRF [14–18]. It is evident that the pulverization and thorough mixing of large ceramic fragments provide a homogeneous

representative sample for further elemental analysis. However, such an approach is inapplicable to valuable artifacts. In this instance, to minimize damage to the original artifact, cutting or drilling can be used to extract a small amount of sample [6,19]. When such a sample preparation procedure is followed by the bulk elemental analysis, it raises the issue of the representativeness of a small subsample removed from a larger ceramic body, especially for coarse pottery.

The main sources of compositional variability of archaeological ceramics [20–22] are "natural" variance due to raw material compositions and manufacturing process ($S_N$), sampling variance ($S_S$) introduced by the sample selection, and the variance introduced in the analytical process ($S_A$):

$$S_T{}^2 = S_N{}^2 + S_S{}^2 + S_A{}^2 \tag{1}$$

The last two terms of Equation (1) are commonly not of interest during ceramic provenance studies. However, these variances can affect the interpretations of archaeological ceramic compositional data and should be taken into account. The analytical variance is the most controllable source of uncertainty in ceramic analysis [21]. It should be evaluated for each analytical protocol, considering the equipment used for measurements, specimen preparation procedure and the element concentration ranges. The sampling variance depends on the sample mass, structure of ceramic matrix, and presence of inclusions of various size; therefore, it should be evaluated for different ceramic types and sampling strategies (cutting, drilling, etc.). However, in routine practice, considering the limited number and size of sherds available for destructive analysis, it is difficult to implement comprehensive estimation of the contributions of both uncertainties due to the sampling and analysis. For geological materials, theoretic modeling using Poisson statistics can be applied to predict the sampling error [23–25]. To our knowledge, the empirical and theoretical assessment of sampling error in the elemental analysis of archaeological ceramics is not common.

## 2. Research Aim

This study describes the assessment of uncertainties introduced at different stages of the ceramic elemental analysis, including the sampling stage (cutting a subsample from the ceramic sherd), preparation of specimens and their measurements by wavelength-dispersive X-ray fluorescence (WDXRF) and ICP-MS methods. Fragments of Neolithic coarse-grained ceramic made by poor mixing of heterogeneous raw materials were used to illustrate a proposed workflow to the assessment of uncertainties. We have addressed the following questions: (i) What is the contribution of the sample heterogeneity to the total uncertainty of the ceramic elemental analysis? (ii) How does the sampling error depend on the distribution of elements within the subsample? (iii) Can sampling errors be described using theoretical modeling?

## 3. Materials and Methods

### 3.1. Object of Study

To date, our research group has carried out studies of the mineral and elemental compositions of pottery from the Popovsky Lug archaeological site located in the valleys of Upper Lena River, Eastern Siberia, Russia [14,17,18,26]. This is a multilayered site containing cultural deposits of various Neolithic stages [27]. The ceramic samples have been previously studied using archaeological classification and petrographic examination. The investigated pottery is dated to the Neolithic period according to relative chronology [28]. Ceramic fragments belong to the pottery of the Ust-Belskaya (Late Neolithic) and Posolskaya (Middle Neolithic) types. In the 1980s, the term "ceramic layer" was introduced [29,30] for the cultural–chronological scheme for the Neolithic of the Baikal region. Later, within the framework of the "ceramic layers", the allocation of "ceramic types" (Posolskaya ceramic type, Ust-Belskaya ceramic type) was created. Some of the Popovsky Lug site's sherds grouped conditionally had a smooth surface and were called "Gladkostennaya". It is quite difficult to determine whether these smooth-walled ceram-

ics belong to a specific typological group without a reconstruction of the vessel. At the Popovsky Lug site, most of the pottery pieces belong to this group.

According to the petrographic analysis of thin sections [26], the mineral composition of different ceramic types from this region is very similar and varies in the following ranges: clay (73–93%), quartz (5–20%) with a grain size of 3 to 0.1 mm, feldspar (1–14%) with a grain size of 1–0.1 mm, and mica (no more than 1%). Other minerals in thin sections include calcite, single inclusions of epidote, ilmenite, hematite, magnetite, limonite, and zircon. In addition, grog (argillaceous clay fragments) and rock fragments identified as granite, chert, sandstone, quartzite, are observed in the thin sections. An example of a thin section, which demonstrates that ceramic samples from the Popovsky Lug archaeological site have inclusion-rich clay matrix and a very heterogenic composition, is shown in Figure 1.

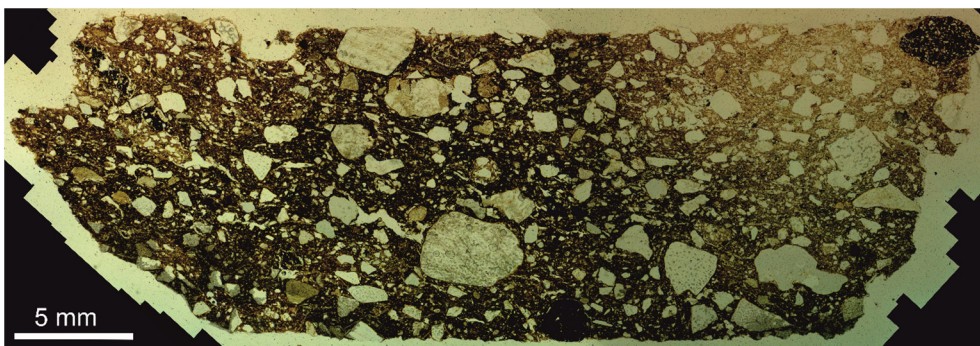

**Figure 1.** Photo of a thin section of pottery from the Popovsky Lug archaeological site.

The mass of most ceramic sherds from the Popovsky Lug archaeological site is relatively small (typically in the range of 2–5 g), and we could use only a small part of the sample for the elemental analysis. Our sampling strategy for elemental analysis was based on cutting a subsample of about 250–350 mg from the original sherd, milling it to powder, and analyzing it using spectral methods. Taking into account the heterogeneity of the studied ceramics, we decided to assess the variability of the results of the elemental analysis due to the sampling and analysis procedures of the example of four fragments, no. 9, 62, 63, and 66.

### 3.2. Description and Preliminary Preparation of Ceramic Samples for Uncertainty Assessment

Samples no. 9 and 62 belong to the Gladkostennaya group; the outer surface of the sherd is also smooth. Sample no. 63 belongs to the Ust-Belskaya ceramic type; the outer surface has semicircular impression marks, arranged in rows. Sample no. 66 belongs to the Posolskaya ceramic; the outer surface of this sample was decorated by impression with a ribbed spatula. The inner surface of all ceramic fragments is smooth. Photos of the original samples (sherds) and their cross-sections are presented in Figure 2.

The ceramic fragments were washed in distilled water in an ultrasonic bath for 1 h at a temperature of about 50 °C and dried in an oven at 80 °C. The fragment of Ceramic no. 9 was ground as a whole in an agate mortar to a particle size less than 50 μm. Grinding the ceramic samples in an agate mortar was found to be a suitable technique with minimal contamination effects [19]. The Ceramic Fragments no. 62, 63, and 66 were cut into separate subsamples that were about 5 mm wide, using an abrasive diamond cutting disk (see the scheme in Figure 2). The number (*n*) of the subsamples depended on the size of the initial sherd (*n* = 8 for Samples no. 62 and 63, *n* = 11 for Sample no. 66). A part of the middle subsample was kept for non-destructive measurements. The rest of the ceramic subsamples were independently ground in an agate mortar by hand. The mass of most of the subsample powder was 250–350 mg in average after grinding.

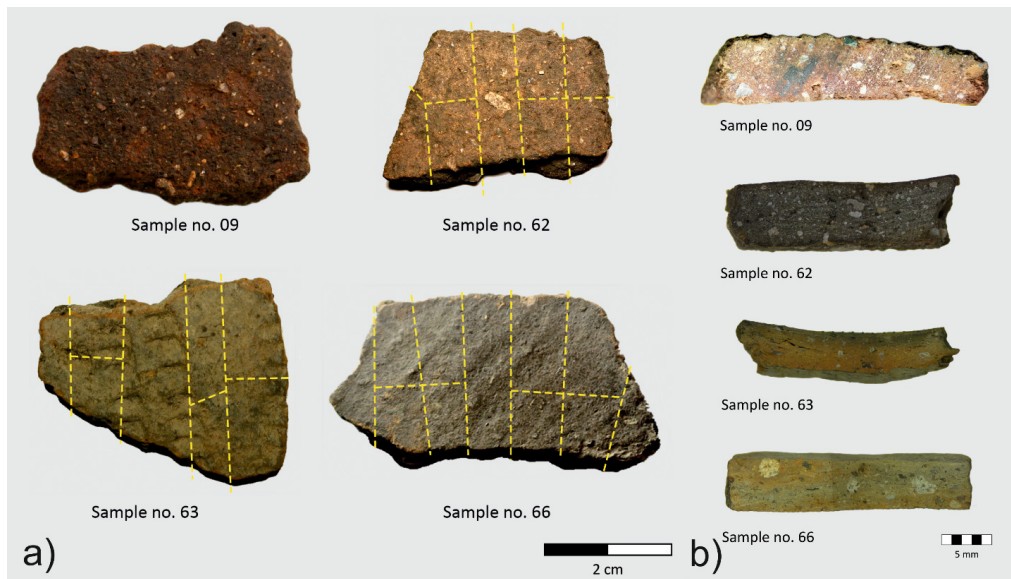

**Figure 2.** Photos of the original ceramic sherds (**a**) and their cross-sections (**b**). The dotted lines are cutting scheme.

*3.3. Methods*

- WDXRF
  WDXRF measurements were performed with a wavelength-dispersive XRF spectrometer S8 Tiger (Bruker AXS, Karlsruhe, Germany) equipped with the Rh anode X-ray tube, and an 8 mm collimator mask for the measurement of small samples. The powdered samples weighing 150 mg were dried at 950 °C for 4 h and the loss on ignition (LOI) values were determined. Then, a mixture of calcined sample weighing 110 mg, 1.1 g of extra-pure lithium metaborate and 7 drops of 40 mg/mL LiBr solution was fused in a platinum crucible in the electric furnace TheOX (Claisse, Québec, QC, Canada) at 1050 °C for 19 min to prepare glass disks with a diameter of 10–12 mm [31]. This technique was previously successfully applied to the elemental XRF analysis of bottom sediments and ancient ceramics [14,32]. Certified reference materials (CRMs) were used to construct calibration curves and control the accuracy, namely silts (BIL-1, BIL-2, SGH-1, SGH-3, SGH-5), loose sediments (SGHM-1, SGHM-2, SGHM-3, SGHM-4), aleurolite (SA-1), clays, slits, and ooze (SDO-1, SDO-2, SDO-8, SDO-9), provided by the Vinogradov Institute of Geochemistry and Research Institute of Applied Physics from Russia; CRMs of sedimentary rocks (JSD-1, JSD-2, JSD-3, JLK-1) were provided by the Geological Survey of Japan; CRMs CH-1 (marine sediment, GeoPT-10), UoK Loess (Köln loess, GeoPT-13), SdAR-1 (modified river sediment, GeoPT-31), and DBC-1 (clay, GeoPT-33) were provided by the International Association of Geoanalysts;

- ICP-MS
  ICP-MS analysis was performed with an Agilent 7500ce quadrupole mass spectrometer (Agilent Technologies Inc., USA). The preparation of the sample was carried out as follows: 100 mg of powdered and dried sample was carefully mixed with 400 mg of lithium metaborate in a 30 mL glassy carbon crucible. The samples were fused in a muffle furnace at 1100 °C for 7 min. After the bead cooled, 3 mL of HF and 1 mL of $HNO_3$ were added and allowed to stand overnight at room temperature. The next day, the mixture was evaporated to dryness. Then, 30 mL of 4 vol.% $HNO_3$ was added to the residue. The solution was stirred with a magnetic stirrer for the complete dissolution of the bead. The resulting solution was filtered into 100 mL volumetric flasks and diluted to volume with 4 vol.% $HNO_3$. Then, 0.5 mL aliquot was transferred to a 15 mL polypropylene test tube and diluted with 9.5 mL of 2 vol.% $HNO_3$. The final dilution factor was 20,000. Before analysis, 100 µL of In solution (10 ng/mL) and

100 µL of Bi solution (10 ng/mL) were added to 10 mL of a sample in accordance with internal standards. Calibration curves were constructed, and accuracy testing was carried out using the geological reference materials of sedimentary, ultramafic, and mafic rocks. This technique was previously successfully applied for the elemental analysis of rocks and sediments [33,34];

- Scanning electron microscopy (SEM-EDS)
  The scanning electron microscope MIRA 3 LMH (Tescan, Czech Republic) was used for the investigation of mineral composition. Epoxy-mounted and polished samples were coated with a thin carbon film (thickness 20–30 nm) to remove any electric charge, applying the Q150R ES vacuum coater (Quorum Technologies, Lewes, UK). The elemental composition of the base matrix and inclusion mineral phases was determined by the AztecLive Advanced Ultim Max 40 microanalysis system with a nitrogen-free energy dispersive spectrometer (EDS) (Oxford Instruments Analytical Ltd., High Wycombe, UK) that allows the simultaneous recording of the intensity of the X-ray spectrum of all elements. SEM-EDS analysis was performed at an acceleration voltage of 20 kV, a beam intensity pulse of 18.50 pulse, an absorbed current of 1.6 nA, and beam diameter of 33 nm;

- µXRF analysis
  Micro-X-ray fluorescence analysis performed by a Tornado M4+ spectrometer (Bruker Nano, Berlin, Germany) was used for elemental mapping. A spectrometer was equipped with an X-ray tube with Rh anode and polycapillary focusing optics. Area mapping was carried out with 20 µm pixel size and 50 ms dwell time; the sample chamber was kept at 25 mbar vacuum. Mapping was carried out in two sequential runs (so-called frames); the final map is a sum of two frames. The samples were epoxy-mounted and polished prior to the analysis.

## 4. Results and Discussion

### 4.1. An Experimental Workflow for Estimating the Uncertainties

To perform a multivariate analysis of variance, we should analyze an extensive set of relatively large ceramic samples belonging to one production site. Given the experimental constraints (sample sizes and limited assemblage), we have decided to carry out an evaluation of variance by means of a detailed study of four samples; see Figure 3 for the workflow scheme. Table S1 (Supplementary Materials) contains the average composition of Ceramic Samples no. 9, 62, 63, 66, obtained by WDXRF (major oxides, wt.%) and ICP-MS (minor and trace elements, µg/g).

To evaluate the uncertainty introduced by the sampling strategy based on cutting a small subsample from the original sherd, we chose three samples, Samples no. 62, 63, and 66, from different ceramic groups. A preliminary study of thin sections under the polarizing microscope showed a heterogeneous ceramic composition and many aplastic components, mainly presented by medium- to fine-grained quartz, feldspar, mica, accessory minerals, fragments of coarse rocks, grog (Figure S1, Supplementary Materials). For the detailed study of the composition of the mineral phases and the elemental distribution across the ceramic section, a ceramic cross-section was studied using SEM and µXRF methods. Both methods also give semiquantitative data on the element concentrations, but in this study, we used them due to their imaging capacities for the qualitative study of the ceramic elemental heterogeneity.

Separately milled subsamples extracted from Samples no. 62, 63, and 66 were analyzed by WDXRF and ICP-MS. The limited quantity of the extracted powdered subsamples does not allow for the reliable assessment of the precision of the analytical method ($S_A$), so this variance was evaluated using Sample no. 9, which was powdered as a whole for the replicate sample preparation and multiple measurements.

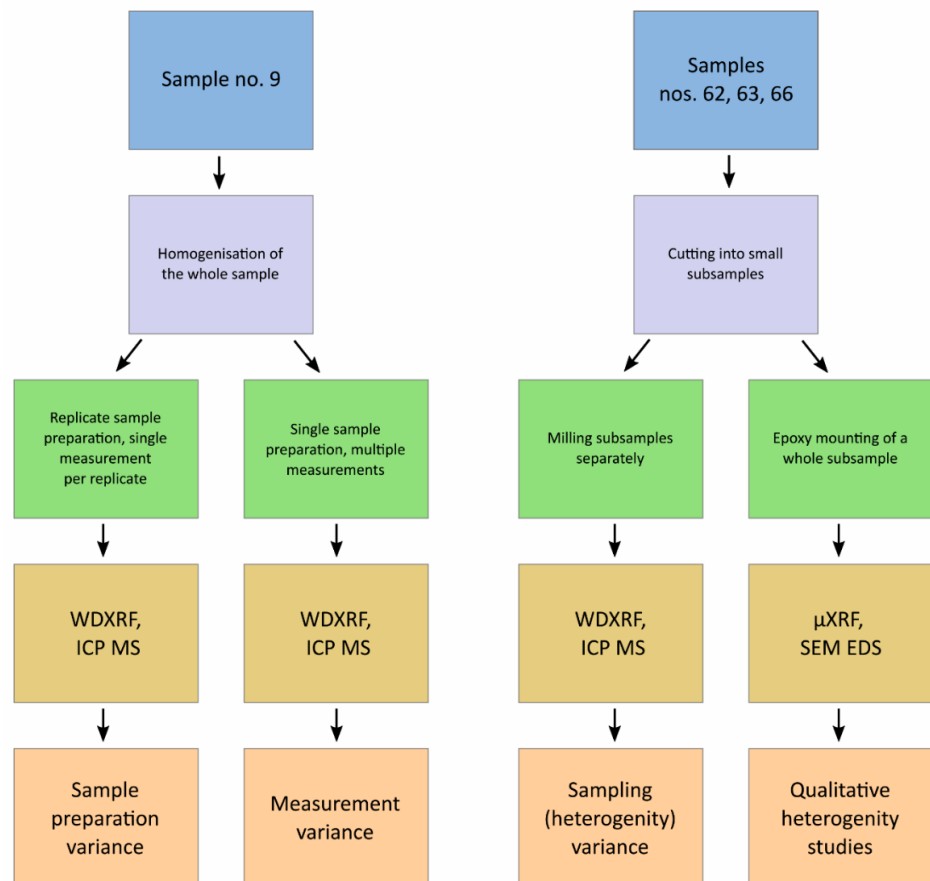

**Figure 3.** Workflow to study the elemental variability of archaeological ceramics.

### 4.2. Evaluation of Uncertainties Introduced by WDXRF and ICP-MS Analysis

The variance introduced in the analytical process ($S_A$) depends primarily on the stability of the instrument's measurement ($S_{meas}$) and the sample preparation procedure ($S_{prep}$): $S_A^2 = S_{meas}^2 + S_{prep}^2$. Both WDXRF and ICP-MS techniques based on the fusion of ceramic powder with borate fluxes demonstrate that there is a possibility to prepare a homogeneous specimen, avoiding the mineralogical and granulometric effects. We did not consider elements with concentrations close to the limits of quantification and elements that had non-stable measuring signals due to counting statistics and sample preparation (e.g., V, Cr, Co, Ni, Zn for WDXRF and Si, Ge, Mo, Sn, W, Tl, Ta for ICP-MS). Therefore, we can assume that the magnitude of the $S_A$ value should be minimal.

Five independent replicates from Sample no. 9 were prepared in accordance with the sample preparation procedures described above, and measured once to assess the coefficient of variation (the relative standard deviation as a percent of the mean value) that characterized the sample preparation uncertainty (coefficient of variation, $CV_{prep}$). In addition to that, one of the replicates was measured five times to assess the coefficient of variation that characterized the measurement uncertainty (coefficient of variation, $CV_{meas}$). Table 1 contains the calculated $CV_{meas}$, $CV_{prep}$, $CV_A$ values. The major elements are presented as oxides.

**Table 1.** Uncertainties of measurement ($CV_{meas}$), sample preparation ($CV_{prep}$), and total analytical procedure ($CV_A$) for WDXRF and ICP-MS analyses of ceramic, %.

| Compound | WDXRF | | | ICP-MS | | |
|---|---|---|---|---|---|---|
| | $CV_{meas}$ | $CV_{prep}$ | $CV_A$ | $CV_{meas}$ | $CV_{prep}$ | $CV_A$ |
| $Na_2O$ | 2.0 | 1.8 | 2.7 | 1.8 | 1.3 | 2.2 |
| $MgO$ | 2.0 | 1.7 | 2.6 | 0.62 | 3.0 | 3.0 |
| $Al_2O_3$ | 0.47 | 0.47 | 0.66 | 0.38 | 1.5 | 1.6 |
| $SiO_2$ | 0.11 | 0.56 | 0.57 | - | - | - |
| $P_2O_5$ | 2.7 | 1.5 | 3.1 | 1.0 | 2.1 | 2.3 |
| $K_2O$ | 0.60 | 0.78 | 1.0 | 0.64 | 1.1 | 1.3 |
| $CaO$ | 0.54 | 0.90 | 1.1 | 0.79 | 6.8 | 6.9 |
| $TiO_2$ | 1.7 | 0.89 | 1.9 | 0.47 | 1.2 | 1.3 |
| $MnO$ | 1.0 | 1.2 | 1.6 | 0.58 | 3.0 | 3.0 |
| $Fe_2O_3$ | 0.18 | 0.62 | 0.64 | 0.42 | 1.4 | 1.4 |
| V | - | - | - | 0.30 | 1.9 | 2.0 |
| Cr | - | - | - | 0.71 | 4.3 | 4.3 |
| Ni | - | - | - | 2.5 | 6.4 | 6.9 |
| Cu | - | - | - | 6.1 | 8.6 | 11 |
| Zn | - | - | - | 0.41 | 7.5 | 7.5 |
| Ga | - | - | - | 0.54 | 1.7 | 1.8 |
| Rb | - | - | - | 0.32 | 2.7 | 2.7 |
| Sr | 7.9 | 7.6 | 11 | 0.35 | 7.8 | 7.8 |
| Y | - | - | - | 0.38 | 2.2 | 2.2 |
| Zr | 2.2 | 5.8 | 6.2 | 0.20 | 5.6 | 5.6 |
| Ba | 6.0 | 6.5 | 8.8 | 0.33 | 2.3 | 2.3 |
| La | - | - | - | 0.44 | 5.9 | 5.9 |
| Ce | - | - | - | 0.38 | 3.6 | 3.7 |
| Nd | - | - | - | 0.83 | 5.2 | 5.3 |
| Th | - | - | - | 1.0 | 4.0 | 4.1 |
| U | - | - | - | 0.78 | 2.7 | 2.9 |

-: non-determined.

As can be seen in Table 1, for the WDXRF method, the estimated $CV_A$ values are satisfactory for all major oxides and less than 3%. For $P_2O_5$ and $TiO_2$, the main contribution to $CV_A$ is made by the measurement uncertainty ($CV_{meas}$) due to their low contents and counting statistics. For minor elements (Sr, Zr, and Ba), $CV_A$ does not exceed 11%.

The estimated analytical precision of the ICP-MS analysis is also satisfactory for all elements considered, with $CV_A$, in general, below 6% except for Ca, Ni, Cu, Zn, and Sr for which the $CV_A$ is higher but does not exceed 11%. As is well-known, ICP-MS provides good precision for rare-earth elements (REEs). We did not list all REEs (only Y, La, Ce, Nd) in Table 1 because for other REEs, the estimated analytical uncertainty of the ICP-MS analysis was also less than 6%.

The measurement uncertainty of ICP-MS is very low in most cases (less than 1%). If we compare the $CV_A$ values for minor elements (e.g., Sr and Zr) obtained by different methods, we can see that the measurement uncertainty for WDXRF is much higher than for ICP-MS due to the counting statistics, but the sample preparation uncertainties are very close for both methods.

According to the IAEA-CU-2006-06 PROFICIENCY TEST [35], the limit of acceptable precision of the analytical method for the determination of major, minor, and trace elements in ancient Chinese ceramic is 20–25 rel.%. Therefore, we conclude that both methods provide the precision required for all the analytes considered.

*4.3. Evaluation of Uncertainty Introduced by Sampling (Heterogeneity)*

Table 2 contains the variation coefficient ($CV_S$), calculated as deviations between analytical results for independent subsamples prepared from one sherd.

**Table 2.** Sampling uncertainties ($CV_S$, %) introduced by ceramic heterogeneity.

| Compound | No. 62 | | No. 63 | | No. 66 | |
|---|---|---|---|---|---|---|
| | WDXRF | ICP-MS | WDXRF | ICP-MS | WDXRF | ICP-MS |
| $Na_2O$ | 4.1 | 3.2 | 3.1 | 3.8 | 14 | 4.8 |
| $MgO$ | 5.7 | 4.4 | 1.5 | 4.0 | 1.9 | 1.9 |
| $Al_2O_3$ | 3.1 | 3.5 | 3.9 | 3.5 | 0.7 | 1.8 |
| $SiO_2$ | 1.4 | - | 1.9 | - | 0.52 | - |
| $P_2O_5$ | 17 | 14 | 15 | 16 | 5.0 | 7.4 |
| $K_2O$ | 5.5 | 6.2 | 4.7 | 4.7 | 1.4 | 2.7 |
| $CaO$ | 2.5 | 4.2 | 5.9 | 9.8 | 16 | 18 |
| $TiO_2$ | 3.4 | 4.2 | 2.9 | 2.4 | 1.1 | 2.3 |
| $MnO$ | 14 | 14 | 51 | 47 | 7.9 | 10 |
| $Fe_2O_3$ | 13 | 14 | 2.8 | 3.2 | 1.6 | 2.7 |
| V | - | 4.9 | - | 4.7 | - | 2.6 |
| Cr | - | 7.1 | - | 2.2 | - | 4.7 |
| Ni | - | 9.4 | - | 6.8 | - | 7.7 |
| Cu | - | 10 | - | 12 | - | 5.5 |
| Zn | - | 20 | - | 12 | - | 9.3 |
| Ga | - | 3.5 | - | 4.5 | - | 2.0 |
| Rb | - | 6.1 | - | 6.7 | - | 2.2 |
| Sr | 7.9 | 9.2 | 6.7 | 7.9 | 12 | 8.4 |
| Y | - | 8.8 | - | 2.5 | | 2.3 |
| Zr | 8.6 | 7.6 | 7.2 | 6.3 | 4.7 | 2.7 |
| Ba | 11 | 8.9 | 7.4 | 8.1 | 5.8 | 3.3 |
| La | - | 6.2 | - | 2.7 | - | 2.3 |
| Ce | - | 4.0 | - | 5.7 | - | 2.1 |
| Nd | - | 8.1 | - | 3.6 | - | 3.7 |
| Th | - | 5.7 | - | 2.6 | - | 1.7 |
| U | - | 6.3 | - | 5.6 | - | 3.9 |

-: non-determined.

We can see in Figure 4, which uses Sample no. 62 as an example, that the $CV_A$ and $CV_S$ values obtained by WDXRF (major oxides) and ICP-MS (minor and trace elements) are compared. It can be seen that for this sample, the uncertainties due to sampling are higher compared to the uncertainties due to the analytical process for most analytes.

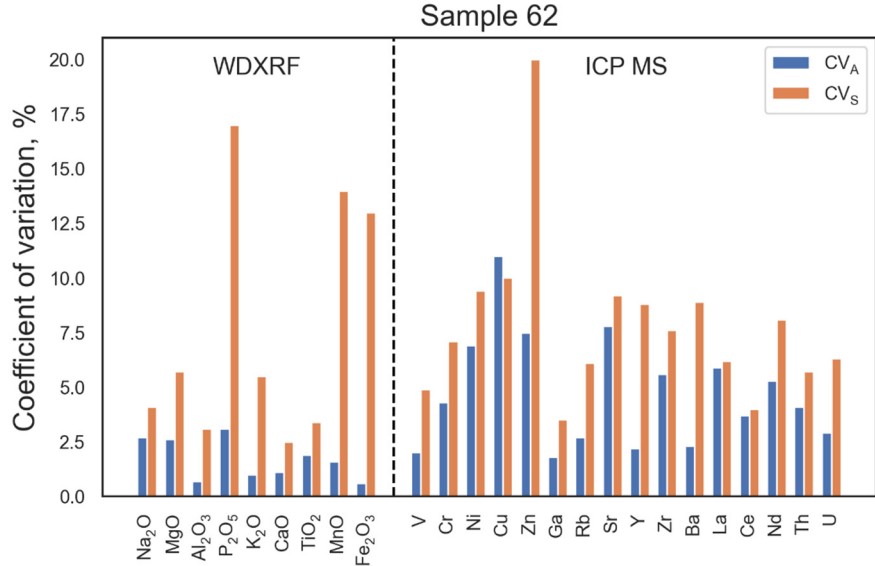

**Figure 4.** Comparison of uncertainties due to the analytical process ($CV_A$) and the ceramic heterogeneity ($CV_S$).

As can be seen in Table 2, $CV_S$ values depend on the sample analyzed. Most of the major oxides are distributed relatively homogeneously within the fragment. The high uncertainty of $Na_2O$ (14% for Sample no. 66) obtained by WDXRF compared to ICP-MS can be explained by its low content (0.3 wt.%), which is close to the WDXRF quantification limit. Other samples (nos. 9, 62, 63) contain approximately 1 wt.% $Na_2O$ (Table S1). Both methods showed high variations (>14–15%) for $P_2O_5$, CaO and MnO (no. 63). The similarity between the $CV_S$ values for WDXRF and ICP-MS is a good indication of an adequate estimate of the sampling variance of these elements.

Figure 5 visually demonstrates the relative deviations of the concentration for individual subsamples ($C_i$) from the average concentration between all subsamples ($C_{mean}$) obtained by WDXRF: $(C_i - C_{mean})/C_{mean} \cdot 100\%$. This figure explains the high variations in $P_2O_5$ (no. 62 and no. 63), CaO (no. 66), MnO (no. 63), and $Fe_2O_3$ (no. 62), seen in Table 2.

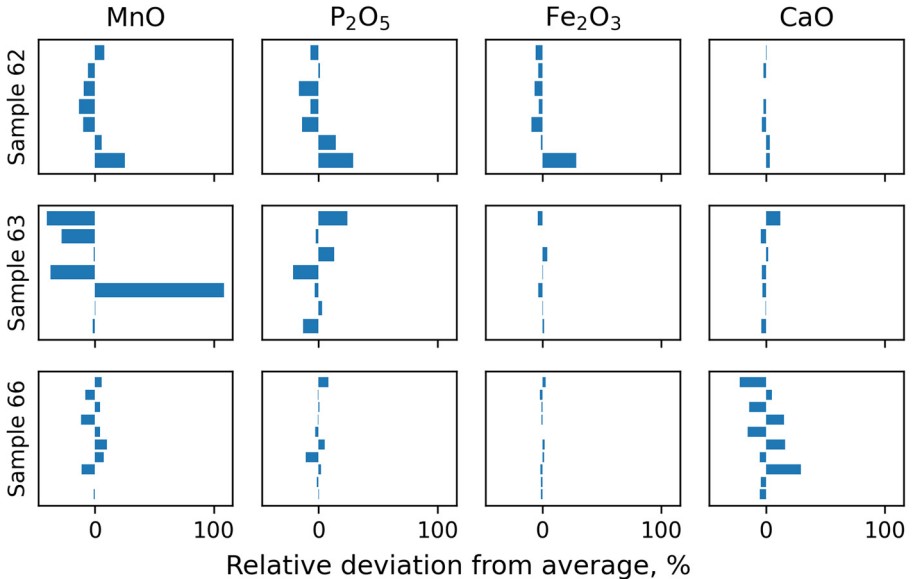

**Figure 5.** Relative deviation of the concentration obtained for individual subsamples from the mean concentration in the sample.

Taking into account the uncertainty values for trace elements (Table 2), we can conclude that our sampling strategy provides the determination of elemental concentrations that vary between subsamples in narrow ranges ($CV_S < 12\%$) except for Zn ($CV_S = 20\%$) for Sample no. 62. Variations of all REEs, which are of particular importance in defining compositional groups [3], were less than 10%.

In summary, these results show that 250–350 mg of ceramic is sufficient to obtain good reproducibility (2–15%) between separate subsamples for most major and trace elements, even for heterogeneous Neolithic ceramics. To further clarify the high variations of $P_2O_5$, CaO, and MnO due to the sampling, we have studied the cross-section of the ceramic samples by SEM and μXRF.

### 4.4. The Heterogeneity Characterization of Ceramic Cross-Section by SEM and μXRF

The SEM investigations showed that the size of the mineral phases in the ceramic cross-sections ranges from 3–5 to 1000–1500 μm. Individual large gray and light gray silicate phases, represented by minerals, such as quartz, feldspars, biotite, and chlorite, are distinguished along the matrix in all samples studied. Additionally, light and bright ore phases, represented mainly by magnetite, rutile, and ilmenite, are observed. Single zircons are found. The elemental composition of the clay matrix is represented by an aluminosilicate component (O, Na, Mg, Al, Si, P, S, Cl, K, Ca, Ti, Mn, and Fe). Figure 6 shows examples of the mineral phases found for Samples no. 62, 63, and 66.

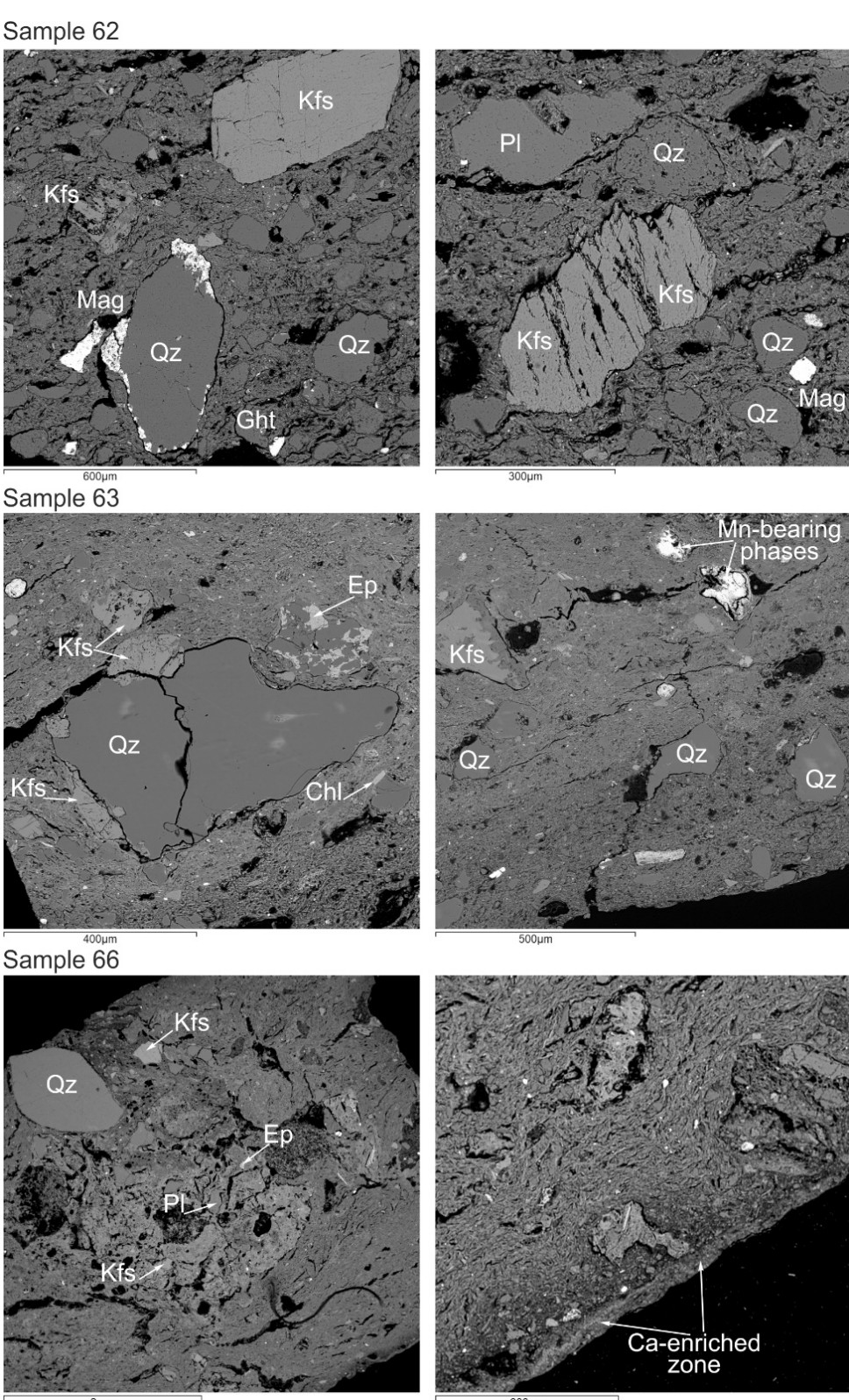

**Figure 6.** The scanning images in back scattered electrons with examples of mineral phases (Kfs—K-feldspar; Qz—quartz; Mag—magnetite; Ght—goethite; Pl—plagioclase; Ep—epidote; Chl—chlorite). The compositions of the mineral phases determined by the SEM-EDS method are presented in Table S2, Supplementary materials.

Some individual features of the composition can be distinguished for each sample. Many iron-rich inclusions (magnetite and goethite) were found in Sample no. 62. This may explain the relatively high sampling variation of $Fe_2O_3$ in this sample compared to other samples. Sample no. 63 contains phases with a high manganese content, which causes a high sampling variation for MnO in this sample, but we were unable to reliably identify a particular Mn-rich mineral. For Sample no. 66 the increased calcium content is observed in the marginal zones of the sample, as opposed to the matrix composition in the central parts, which is probably related to the post-depositional processes.

Both Samples no. 62 and no. 63 contain many inclusions of feldspar in the clay matrix. For these samples, the sampling uncertainties are larger for Si, Al, K, and Ba (see Table 2), which are part of this mineral, compared to Sample no. 66. What causes the heterogeneity of the distribution of P for Samples no. 62 and no. 63 is difficult to explain. According to the SEM data, the content of $P_2O_5$ varies greatly in both the clay matrix and the inclusions.

The μXRF method allows obtaining maps of distribution of elements. Maps for selected elements (Ca, K, and Si) are presented in Figure 7. It should be noted that the obtained μXRF maps are in good agreement with the SEM results. The data of the SEM study indicate that there are many phases of quartz and feldspar in the ceramic paste for all three studied fragments. This is clearly visualized in the Si and K μXRF maps, respectively.

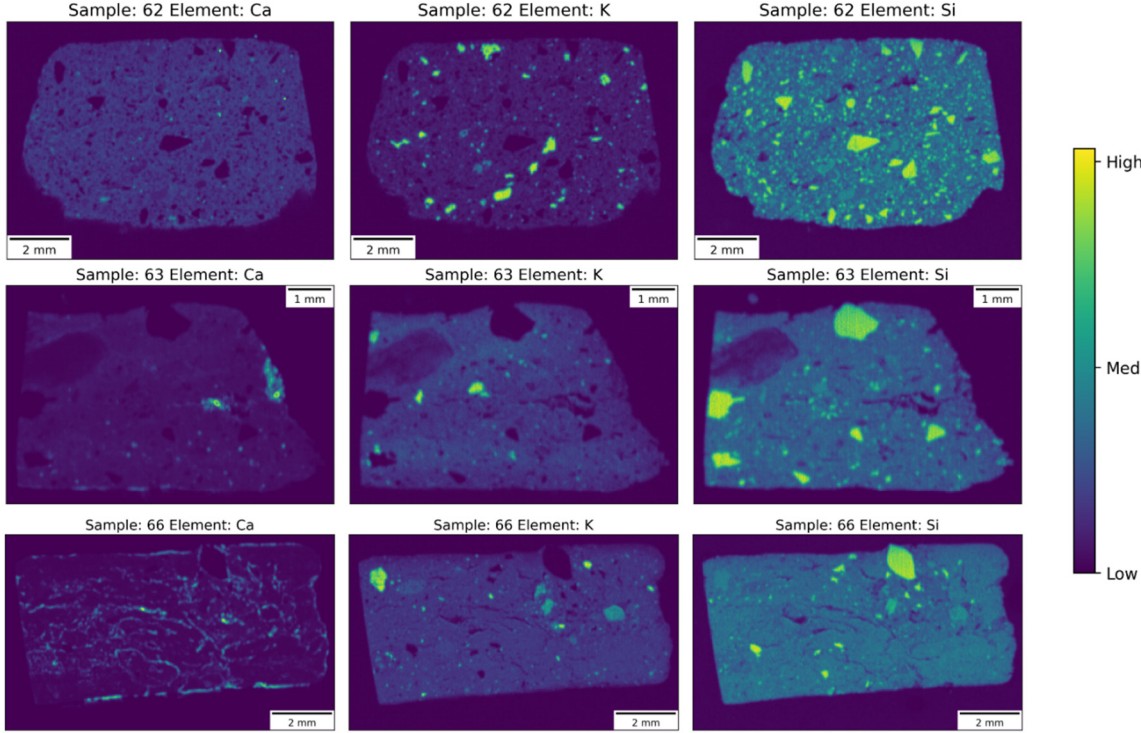

**Figure 7.** μXRF elemental distribution maps for Samples no. 62, 63, 66. Relative intensity scales used.

The calcium distribution was found to be significantly different among the samples studied. Samples no. 63 and 66 are relatively more heterogeneous than Sample no. 62, which is also supported by the SEM data. Ca-enriched zones were observed for both the external surfaces and the pore surfaces of Sample no. 66, which may be an indication of post-deposition alteration of Ca. Moreover, CaO is a significant part of the chemical composition of the raw materials used in the ceramic production, as the studied archaeological site is located in the field of distribution of Ca-rich rock. Therefore, the presence of Ca-enriched zones in the sherds can be due not only to the post-depositional chemical alteration, but also due to the extraction of calcium from the ceramic paste compounds during long-term burial.

Maps of Mn and P for Sample no. 63 are presented in Figure 8. This sample is characterized by relatively high contents of MnO and $P_2O_5$ (see supplementary Table S1 for WDXRF data). The SEM data for Sample no. 63 indicate that there are large singular Mn-rich ore inclusions. The µXRF map for Mn for Sample no. 63 confirms this observation. This might be the cause for the high heterogeneity variance for Mn for this sample (see Table 2). Variations of P, which are probably related to both mineral inclusions and post-depositional alteration, are also observed on the map. As is well known, phosphorous oxide plays an important role in the alteration of archaeological ceramics during burial [36,37].

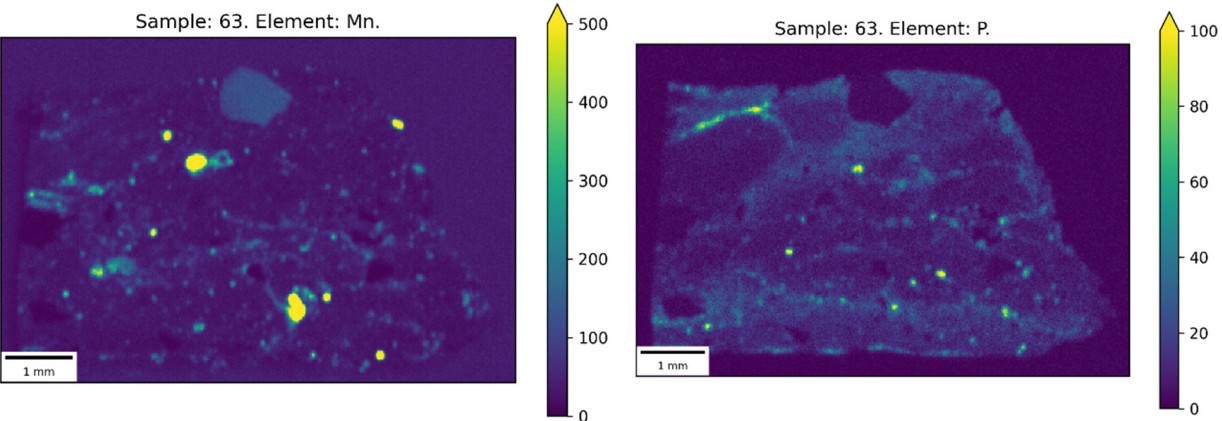

**Figure 8.** µXRF maps of the Mn and P distribution for Sample no. 63. The intensity scales were trimmed to enhance weak details. For full intensity range images, please see Supplementary Materials.

*4.5. Theoretical Modeling of Sampling Error*

The results of the SEM and µXRF study, as well as of the petrographic examination showed the presence of a large inclusions of quartz ($SiO_2$) in the clay matrix, the size range of which is mainly 0.1 to 1 mm; single large inclusions of 2–3 mm in size are less common. Despite the visible heterogeneity in the silicon distribution in the sample, the sampling error for the bulk analysis of $SiO_2$ is less than 2% (Table 2). At the same time, inclusions of manganese lead to a sampling error of up to 50%. To explain the observed differences in the sampling errors, we used the Poisson distribution considering two cases: (1) Both clay matrix and inclusions contain a high amount of analyte; (2) Analyte is mainly concentrated in inclusions.

The sampling error can be theoretically modelled using data on the size of inclusions and their content in the clay matrix. According to the Poisson distribution [23], the relative standard deviation (*RSD*) of the number (*N*) of particles (in our case, the number of inclusions) in the unit volume of the sample matrix is proportional to $RSD = \sqrt{N}/N$. The number of quartz particles (*N*) can be calculated from the mass of a ceramic sample ($M_{cer}$ = 250 mg), the mass fraction of quartz ($\omega_{quartz}$ = 5–20%), the density ($\rho_{quartz}$ = 2.5 g/cm$^3$), and the inclusion size ($D_{quartz}$): $N = (\omega_{quartz} M_{cer}/\rho_{quartz})/D_{quartz}{}^3$. We demonstrated using a model that cubic quartz inclusions with a size (cube edge) $D_{quartz}$ = 100–2000 µm are distributed in a homogeneous clay matrix of the fragment. Silicon is part of both the

clay matrix and quartz; therefore, when homogenizing the entire 200 mg subsample, the analytical signal of $SiO_2$ in ceramics ($A_{SiO2}{}^{cer}$) is equal to the sum of the analytical signals of $SiO_2$ in the clay matrix ($A_{SiO2}{}^{clay}$) and quartz ($A_{SiO2}{}^{quartz}$): $A_{SiO2}{}^{cer} = A_{SiO2}{}^{clay} + A_{SiO2}{}^{quartz}$. The $SiO_2$ content in clay is approximately 60 %, and the $SiO_2$ content in quartz is 100%. The dispersion of $SiO_2$ in the ceramic ($\delta_{SiO2}{}^{cer}$) depends on the dispersion of $SiO_2$ in the clay matrix and the dispersion of $SiO_2$ due to quartz inclusions: $(\delta_{SiO2}{}^{cer})^2 = (\delta_{SiO2}{}^{clay})^2 + (\delta_{SiO2}{}^{quartz})^2$.

We used the approximation that the $SiO_2$ content in the clay matrix is constant and the dispersion of the $SiO_2$ is close to zero ($\delta_{SiO2}{}^{clay} \approx 0$). Therefore, variations in the $SiO_2$ content in a homogenized ceramic fragment are determined only by variations in the number of quartz inclusions in ceramics. Assuming that the analytical signal is proportional to the mass fraction of quartz ($A_{SiO2}{}^{quartz} \sim \omega_{quartz}$), the variance of $SiO_2$ depends on the mass fraction of quartz, and the number of inclusions and can be characterized by the relative standard deviation: $RSD = \omega_{quartz}\sqrt{N}/N$. This approximation is quite rough; it considers only the quartz variations, but does not consider $SiO_2$ content in a clay matrix.

Figure 9 shows the *RSD* values of different inclusions sizes and the quartz mass fraction of a 250 mg sample. Since the main size of the quartz phase is less than 1000 μm, the theoretical value of *RSD* is less than 5%, which does not contradict the experimentally obtained variations due to sampling. Similar conclusions can be drawn for other components that are present in both the clay matrix and the inclusions (for example, $Al_2O_3$ and $K_2O$ in the inclusions of feldspar).

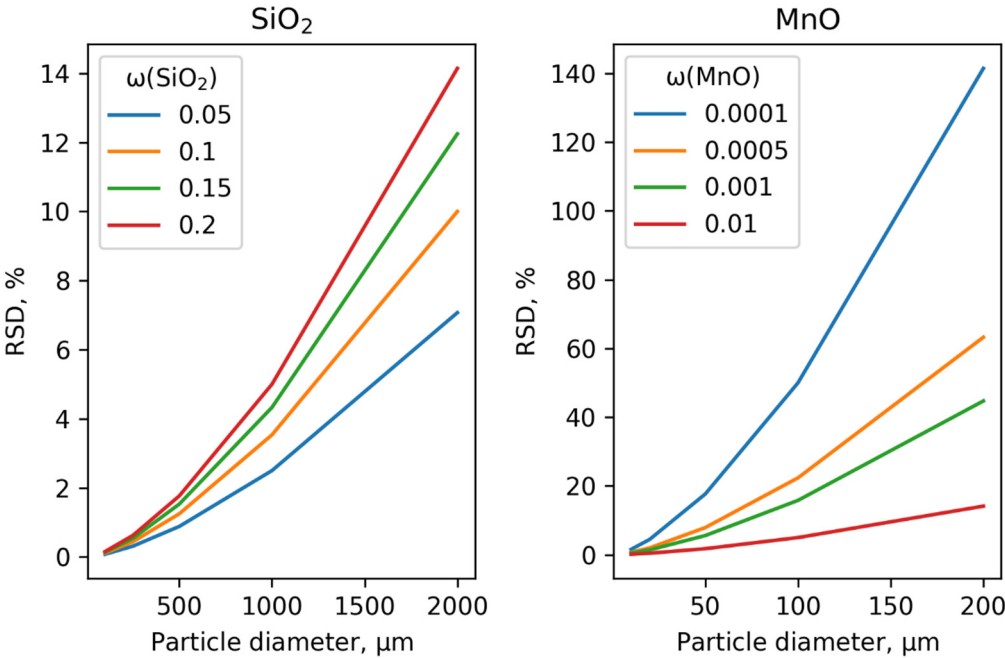

**Figure 9.** Theoretical sampling error for two cases: (1) Both clay matrix and inclusions contain high amount of analyte (Si); (2) Analyte (Mn) is mainly concentrated in inclusions.

Using the Poisson sampling error model for the case of small inclusions of ore minerals in a clay matrix, a different dependence will be observed. For example, for Sample no. 63 (Figures 6 and 8), phases with a high Mn content (up to 80% calculated as oxide MnO) are visible, the size of inclusions of which is about 100 μm. The mass fraction of such inclusions is very small. The content of Mn in the clay matrix is less than 0.2%, calculated as MnO; therefore, unlike the case considered for quartz, the manganese content in the clay will not affect the sampling error. For this type of ore inclusion, the relative standard deviation of the number of inclusions (*N*) per unit volume of the clay matrix is proportional to $RSD = \sqrt{N}/N$.

Figure 9 shows the *RSD* values for the following model: the density of the Mn phases is $\rho_{ore}$ = 5 g/cm$^3$, the mass of the ceramic fragment is $M_{cer}$ = 250 mg, the fraction of inclusions is $\omega_{ore}$ = 0.0001–0.01 wt.%, the size of cubic inclusions distributed in the clay matrix is $D_{ore}$ = 10–200 μm. The analytical signal depends only on the content of the component analyzed in the ore phase. As can be seen in Figure 9, the *RSD* sampling error can exceed 100 % for a small content of large inclusions. This agrees with the experimental data obtained for Sample no. 63, where relatively large inclusions with high Mn content were observed. For this sample, the sampling error reaches 50% due to heterogeneity.

It is clear that the real ceramic samples have no ideal size, shape, and composition of mineral grains. However, such calculations make it possible to qualitatively assess possible sampling errors. Obviously, an increase in the subsample mass will reduce the sampling error.

## 5. Conclusions

The main idea of this study is to show a workflow for the estimation of the uncertainties introduced by sample heterogeneity (i.e., sampling uncertainty) and the analytical process (measurement and sample preparation uncertainty) in the ceramic elemental analysis. Considering the requirement of limiting the number and size of sherds available for destructive analysis, classic multi-factor variance analysis is not possible. So, we used the approach when the uncertainty associated with the ceramic elemental heterogeneity and the uncertainty introduced during the analytical process could be evaluated separately.

The proposed workflow has been successfully applied to the sherds from the archaeological site Popovsky Lug (Baikal Siberia). It was shown that the uncertainties due to measurements and sample preparation for the WDXRF and ICP-MS analyses were satisfactory for the analytes considered and did not exceed 11%. The uncertainties due to the heterogeneity of the sample depended on the type of ceramic and the element under consideration. The sampling strategy (cutting 250–350 mg of subsample) provides low variations (<12%) of most trace elements that are widely used in defining compositional groups. The variance due to the sample heterogeneity for major oxides was acceptable (<15%) in most cases, but it appeared to be greater for phosphorus, calcium, and manganese. The reasons for the high variations in the concentrations of these elements have been explained by measuring the cross-sections of the initial fragments using the SEM and μXRF methods. Additionally, using the Poisson sampling error model allowed us to explain the low and high variations of different elements considering the presence/absence of the analyte as part of the clay matrix and specific minerals.

Evidently, the observed elemental variations are specific characterizations of coarse pottery from the archaeological site Popovsky Lug included in our case study and cannot be generalized to other types of Neolithic sherds of Baikal Siberia. However, the described experimental design can be integrated in other archaeological studies to find uncertainties derived from sample preparation, measurement, and sampling in the elemental analysis when the number and size of ceramic sherds are limited for destructive analysis. Elements with high variations in concentration should be excluded from the evaluation for the reliable interpretations of ceramic elemental composition data.

As sample size is clearly an issue for ancient ceramics, another way to approach uncertainty evaluation is to have made our own large ceramic tiles from clay that is typical of that region, with percent contents of temper and other inclusions of different sizes. This is a potential future project through which we will explore our findings more thoroughly.

**Supplementary Materials:** The following supporting information can be downloaded at: https://www.mdpi.com/article/10.3390/heritage6050234/s1, Figure S1: Dominant mineralogical combinations of the ceramics for our samples (Q—quartz, Mica, Fsp—plagioclase or K-feldspar, Ac—accessory minerals and Rock–fragments of granite and chert, grog—argillaceous clay fragments with or without mineral grains) (a,c,e). Microphotographs of pottery thin sections under plane-polarized light (b,d,f). Large inhomogeneous inclusions in ceramics include fragments of rocks and temper. Clear or translucent minerals of an angular, subangular, and subrounded shape are medium- and fine-grained minerals of quartz and plagioclase or K-feldspar; Table S1: The average composition of Ceramic Samples no. 9, 62, 63, 66 obtained by WDXRF (major oxides) and ICP MS (minor and trace elements); Table S2: The composition of mineral phases in ceramic samples obtained by SEM-EDS (wt.%).

**Author Contributions:** G.V.P.: conceptualization, methodology, investigation, writing—original draft, writing—review and editing, visualization, supervision; M.A.S.: conceptualization, methodology (µXRF), formal analysis (µXRF), investigation (µXRF), writing—original draft, writing—review and editing, visualization; M.M.M.: formal analysis, validation; A.L.F.: software, validation; I.V.A.: formal analysis (µXRF), investigation (µXRF); O.Y.B.: formal analysis (SEM data), investigation (SEM data); V.M.C.: formal analysis (WDXRF), investigation (WDXRF), writing—review and editing; A.A.A.: formal analysis (WDXRF), investigation (WDXRF), writing—review and editing; A.S.M.: writing—review and editing, visualization; E.I.D.: resources, conceptualization; D.L.S.: resources. All authors have read and agreed to the published version of the manuscript.

**Funding:** The study was conducted in the frame of the grant of the Russian Science Foundation N 19-78-10084, https://rscf.ru/project/19-78-10084/ (accessed on 21 April 2023).

**Data Availability Statement:** The data obtained are available via the author contacts.

**Acknowledgments:** The research was performed using the equipment of «Geodynamics and Geochronology» center at the Institute of the Earth's Crust SB RAS and «Isotope-Geochemical Studies» center at the A.P. Vinogradov Institute of Geochemistry SB RAS. M. S. and I. A. acknowledge support from M.V. Lomonosov Moscow State University Program of Development for granting access to Tornado M4+ spectrometer.

**Conflicts of Interest:** The authors declare that they have no known competing financial interest or personal relationship that could have appeared to influence the work reported in this paper.

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
