# Peer review of "A Workflow for Uncertainty Assessment in Elemental Analysis of Archaeological Ceramics: A Case Study of Neolithic Coarse Pottery from Eastern Siberia"

_heritage, doi:10.3390/heritage6050234_

Round 1

Reviewer 1 Report

This was a wonderful, well-thought out article that has implications for all types of sampling of ceramic material.  Some minor comments:

1) Sample size is clearly an issue. While I am not asking you to go back and sample more specimens, I do think the argument that sample size was limited due to number and size of sherds available for destructive analysis is faulty.  The obvious solution here is to have made your own ceramic tiles - this way, you could increase sample size, and have control over clay, as well as size, shape, size, and percent of temper.  You should say in your conclusions that this is a potential future project to explore your findings more thoroughly. 

2) Figure 2 caption: explain in the caption what the dotted lines are (cutting scheme). 

Overall, the English is strong. However, there are lots of numerous small grammatical errors and typos that should be fixed. I would say on average, at least one per page.  Two examples: 

1) Page 2, paragraph before Section 2, the sentence: "The sampling variance depends on the sample mass, structure of ceramic, presence of inclusions of various size, therefore it should be evaluated for difference ceramic type and sampling strategy." Grammatically, this sentence is not quite right. One way to correct it would be "The sampling variance depends on the sample mass, structure of ceramic MATRIX, AND presence of inclusions of various size; therefore, it should be evaluated for difference ceramic type and sampling strategy." For proper grammatical use of therefore, see. . . .

https://www.sheaws.com/tip-57-how-do-i-punctuate-therefore/#:~:text=When%20you%20use%20a%20conjunctive,follow%20it%20with%20a%20comma. 

On page 3, Section 3.2, first sentence: "shred" should be "sherd".

Please make sure the entire article is read through for these types of corrections. I circled a total of 9 possible corrections while reading and probably missed some.  

Author Response

This was a wonderful, well-thought out article that has implications for all types of sampling of ceramic material.  Some minor comments:

1) Sample size is clearly an issue. While I am not asking you to go back and sample more specimens, I do think the argument that sample size was limited due to number and size of sherds available for destructive analysis is faulty.  The obvious solution here is to have made your own ceramic tiles - this way, you could increase sample size, and have control over clay, as well as size, shape, size, and percent of temper.  You should say in your conclusions that this is a potential future project to explore your findings more thoroughly. 

We are grateful to the reviewer and added a comment to the text

2) Figure 2 caption: explain in the caption what the dotted lines are (cutting scheme). 

It was corrected

Comments on the Quality of English Language

Overall, the English is strong. However, there are lots of numerous small grammatical errors and typos that should be fixed. I would say on average, at least one per page.  Two examples: 

1) Page 2, paragraph before Section 2, the sentence: "The sampling variance depends on the sample mass, structure of ceramic, presence of inclusions of various size, therefore it should be evaluated for difference ceramic type and sampling strategy." Grammatically, this sentence is not quite right. One way to correct it would be "The sampling variance depends on the sample mass, structure of ceramic MATRIX, AND presence of inclusions of various size; therefore, it should be evaluated for difference ceramic type and sampling strategy." For proper grammatical use of therefore, see. . . .

https://www.sheaws.com/tip-57-how-do-i-punctuate-therefore/#:~:text=When%20you%20use%20a%20conjunctive,follow%20it%20with%20a%20comma. 

On page 3, Section 3.2, first sentence: "shred" should be "sherd".

Please make sure the entire article is read through for these types of corrections. I circled a total of 9 possible corrections while reading and probably missed some.  

We have corrected some errors

Reviewer 2 Report

In this paper,the assessment of uncertainties by sampling error and instrument error in the composition analysis of ancient pottery samples using wavelength-dispersive X-ray fluorescence and inductively

coupled plasma mass spectrometry methods has been studied with an example of the Neolithic pottery sherds from Popovsky Lug. The experimental design is reasonable and discussion of the experimental results is sufficient.

1.Due to the small size of the sample, only 8~11 pieces were cut for relevant testing and analysis, resulting in a relatively small statistical quantity.

2.According to the experimental results, large-sized inclusions in pottery are the main cause of uncertainty in composition analysis. So can we consider sieving out large particle impurities after sampling?

Author Response

1.Due to the small size of the sample, only 8~11 pieces were cut for relevant testing and analysis, resulting in a relatively small statistical quantity.

We agree with the reviewer. In the paper we explain our the experimental constraints (sample sizes, limited assemblage), that did not allow us to expand the number of samples for statistic. As sample size and is clearly an issue for ancient ceramics, another way to the uncertainty evaluation is to have made own large ceramic tiles from typical region clay with percent of temper and other inclusions of different size.  This is a potential future project to explore our findings more thoroughly. Some comments were added to the text

2.According to the experimental results, large-sized inclusions in pottery are the main cause of uncertainty in composition analysis. So can we consider sieving out large particle impurities after sampling?

In our opinion, sieving out large particles will lead to a distortion of the composition of the studied ceramics. Large particles are part of the temper. Their size is a characteristic of pottery itself and the production region.

Reviewer 3 Report

The subject of manuscript sounds interesting and appropriate for the publication in the Heritage journal. In general, the reviewer evaluates the idea and results of this research positively. However, some questions and comments may be suggested.

Terms and definitions.

1. "Ceramic type". What sence is applied to this term? Is is "cultural ceramic type"? If so, two ceramic types are presented in the research - Ust-Belsky type and Posolskaya tytpe according to current archaeological systematization. As for the "Gladkostennaya ceramic type" (samples N09 and 62 - 3.2 sub-section), in current Neolithic systematization of Eastern Siberia there is no this cultural type. Plain-walled, or plain-surfaced, pottery fragments (plain - "gladkii" (гладкий) in Russian) are presented in some amount in the assemblages of Ust-Belsky pottery as well as in Posolskaya pottery assemblages. 

So, the cultural identification of examined samples needs correction.

2. "Neolithic coarse pottery" and "low firing coarse pottery". These definitions need to be clarified and referenced. These characterstics are common for Neolithic pottery of Eastern Siberia as  a whole, or for certain archaeological cultures, or for certain sites? What is the experimental/empiric base of these definitions - research results published earlier, or unpublished authors' investigations?

Archaeological background. 

It may be recommended:

-to add chronology/dating (relative or absolute) for Neolithic Ust-Belsky and Posolskaya cultures;

- to add some precising comments about Popovsky Lug archaeological site. According to V. Vetrov, 2003, this is multilayered site containing cultural deposits of various Neolithic stages;

- to add very short but clear and referenced characteristics of Ust-Belsky and Posolskaya pottery assemblages from Popovsky Lug site. 

Methods. 

1.It is obvious from the manuscript context and suppl. materials that the authors use petrography microscopic study for their research aims. However, in sub-section 3.3. there is no any information about this method. It is not clear, why and how the petrograpgy was applied. 

2. In 3.1 sub-section the petrographic characteristics of Neolithic pottery from research area are noted. The description is too broad, not specific enough. It is unclear, how and at what experimental materials these examination was provided. 

How these results are corresponding to published data on the petrograpgy of East Siberian Neolithic potteries [see - Saviliev N.A., Ulanov I.V. Neolithic Pottery of the Multilayered site Gorelyi Lew (South Angara Region), Bulletin of the Irkutsk State University. Geoarchaeology, Ethnology, and Anthropology Series. 2018, Vol. 26, pp. 46-85; Титова Ю.А. и др. Технологические особенности изготовления керамики неолита и бронзового века стоянки Удачный-14 в г. Красноярске // Известия Лаборатории древних технологий. 2017. Т.13. №3. С. 9-18. 

3. Figure 2 presents cross-sections of examined samples. Obviously, they demonstrate mineralogical and textural diversity and heterogeinity and, probably, belong to different petrographic variants. In what measure results of petrographic examnation are took into account concerning the research program and workflow to study the elemental variability? What is correlation between petrographic variability and elemental variability? 

4.Sub-section 3.2 - SEM-EDS would be soun d more correct that SEM in the context of presented investigation. 

Results and Discussion; Conclusion. 

 it may be recommended to accent practical use of the suggested workflow for the investigations based on evaluating of ceramic elemental analysis data. 

Moderate imroving of the English lanquage is recommended concerning the grammar and stylistic features. 

Author Response

Terms and definitions.

1. "Ceramic type". What sence is applied to this term? Is is "cultural ceramic type"? If so, two ceramic types are presented in the research - Ust-Belsky type and Posolskaya tytpe according to current archaeological systematization. As for the "Gladkostennaya ceramic type" (samples N09 and 62 - 3.2 sub-section), in current Neolithic systematization of Eastern Siberia there is no this cultural type. Plain-walled, or plain-surfaced, pottery fragments (plain - "gladkii" (гладкий) in Russian) are presented in some amount in the assemblages of Ust-Belsky pottery as well as in Posolskaya pottery assemblages. 

So, the cultural identification of examined samples needs correction.

In the 80s. 20th century as a result of the development of a cultural-chronological scheme for the Neolithic of the Baikal region, the term “ceramic layer” was introduced [Berdnikova, 1986; Saveliev, 1989]. Later, within the framework of the "ceramic layers", the allocation of "ceramic types" (Posolsky ceramic type, Ust-Belsky ceramic type) was made. 

References were added to the text.

We corrected information about "Gladkostennaya" ceramic group.

Some of the sherds grouped conditionally had a smooth surface and were called "Gladkostennaya" It is quite difficult to determine whether these plain-walled ceramics belong to a specific typological group without reconstruction of the vessel. At the Popovsky Lug site, this group of pottery is the most numerous.

2. "Neolithic coarse pottery" and "low firing coarse pottery". These definitions need to be clarified and referenced. These characterstics are common for Neolithic pottery of Eastern Siberia as  a whole, or for certain archaeological cultures, or for certain sites? What is the experimental/empiric base of these definitions - research results published earlier, or unpublished authors' investigations?

We took into account the reviewer's comment and decided not to use the term "low firing coarse pottery".

We used the term “coarse pottery” that characterizes that ceramic was made from coarse-grained paste. In its most general form, pottery can be divided into two large groups according to the composition: 1) "rough or coarse" ceramics - with a heterogeneous coarse-grained structure; 2) "fine" ceramics - with a homogeneous fine-grained structure of the shard (porcelain, faience)

Archaeological background. 

In our opinion, within the framework of the presented comprehensive analytical study, a detailed description of the archaeological background is not required.

It may be recommended:

-to add chronology/dating (relative or absolute) for Neolithic Ust-Belsky and Posolskaya cultures;

We added reference on the basis of AMS, the chronology was corrected and proposed for ceramics of the Posolskaya and Ust-Belskaya types of the Upper Lena

Shergin, D.L. Posol’sky-type ceramics of the Upper Lena basin (based on the data of Popovsky Lug and Makarovo I sites). Reports of the Laboratory of Ancient Technologies 2023, 19, 8–32. [in Russian]. DOI: https://doi.org/10.21285/2415-8739-2023-1-8-32.

- to add some precising comments about Popovsky Lug archaeological site. According to V. Vetrov, 2003, this is multilayered site containing cultural deposits of various Neolithic stages;

Information was added.

- to add very short but clear and referenced characteristics of Ust-Belsky and Posolskaya pottery assemblages from Popovsky Lug site. 

The references were added.

Methods. 

1.It is obvious from the manuscript context and suppl. materials that the authors use petrography microscopic study for their research aims. However, in sub-section 3.3. there is no any information about this method. It is not clear, why and how the petrograpgy was applied. 

We added the reference where description of the petrographic examination was presented.

Pashkova, G.V.; Demonterova, E.I.; Chubarov, V.M.; Kaneva E.V.; Shergin, D.L.; Maltsev, A.S.; Amosova, A.A.; Mukhamedova, M.M.; Mikheeva, E.A. Characteristic of the mineral and elemental composition of the Popovsky Lug station (Upper Lena) neolithic ceramics. Proceedings of the Interdisciplinary Archaeological Research of the Ancient Cultures of the Yenisei Siberia and Adjacent Territories; Krasnoyarsk, Russia, 20-21 October 2020, pp. 45–54. [in Russian].

2. In 3.1 sub-section the petrographic characteristics of Neolithic pottery from research area are noted. The description is too broad, not specific enough. It is unclear, how and at what experimental materials these examination was provided. 

We added the reference where description of the petrographic examination was presented.

How these results are corresponding to published data on the petrograpgy of East Siberian Neolithic potteries [see - Saviliev N.A., Ulanov I.V. Neolithic Pottery of the Multilayered site Gorelyi Lew (South Angara Region), Bulletin of the Irkutsk State University. Geoarchaeology, Ethnology, and Anthropology Series. 2018, Vol. 26, pp. 46-85; Титова Ю.А. и др. Технологические особенности изготовления керамики неолита и бронзового века стоянки Удачный-14 в г. Красноярске // Известия Лаборатории древних технологий. 2017. Т.13. №3. С. 9-18. 

We are grateful to the referee for the indicated works. It is difficult to use the petrography data for ceramics from the Gorely Les site for comparison, because 1) the studies were performed on only 8 fragments, 1 of which refers to ceramics, which is not considered within the framework of this article, and the remaining 7 fragments are not clear which ceramics belong to.
As for the data from the Udachny-14 site, they seem to be quite interesting for drawing analogies on petrographic studies of ancient ceramics of Eastern Siberia, which will certainly be taken into account in future work.

3. Figure 2 presents cross-sections of examined samples. Obviously, they demonstrate mineralogical and textural diversity and heterogeinity and, probably, belong to different petrographic variants. In what measure results of petrographic examination are took into account concerning the research program and workflow to study the elemental variability? What is correlation between petrographic variability and elemental variability? 

We used the petrographic examination of thin sections for preliminary study to show a heterogeneous ceramic composition and many aplastic components. We didn't consider the correlation between petrographic variability and elemental variability. For this aim we used SEM and µXRF methods.

4.Sub-section 3.2 - SEM-EDS would be sound more correct that SEM in the context of presented investigation. 

Corrected

Results and Discussion; Conclusion. 

 it may be recommended to accent practical use of the suggested workflow for the investigations based on evaluating of ceramic elemental analysis data. 

The described experimental design can be integrated in other archaeological studies to find uncertainties derived from sample preparation, measurement, and sampling in the elemental analysis when the number and size of of ceramic sherds are limited for destructive analysis.